

# Gram-negative ESKAPE bacteria bloodstream infections in patients during the COVID-19 pandemic

María Dolores Alcántar-Curiel[1,*], Manuel Huerta-Cedeño[1,2,*], Ma Dolores Jarillo-Quijada[1], Catalina Gayosso-Vázquez[1], José Luis Fernández-Vázquez[1], María Luisa Hernández-Medel[3], Manuelita Zavala-Pineda[3], Miguel Ángel Morales-Gil[3], Verónica Alejandra Hernández-Guzmán[3], Manuel Ismael Bolaños-Hernández[3], Silvia Giono-Cerezo[2] and José Ignacio Santos-Preciado[1]

[1] Unidad de Investigación en Medicina Experimental, Facultad de Medicina, Universidad Nacional Autónoma de México, Ciudad de México, Ciudad de México, Mexico

[2] Laboratorio de Bacteriología Médica, Escuela Nacional de Ciencias Biológicas, Instituto Politécnico Nacional, Ciudad de México, Ciudad de México, Mexico.

[3] Unidad de Infectología y Laboratorio Central de Microbiología, Hospital General de México Dr. Eduardo Liceaga, Ciudad de México, Ciudad de México, Mexico

[*] These authors contributed equally to this work.

Corresponding author
María Dolores Alcántar-Curiel,
alcantar@unam.mx

## ABSTRACT

Bloodstream infections due to bacteria are a highly consequential nosocomial occurrences and the organisms responsible for them are usually multidrug-resistant. The aims of this study were to describe the incidence of bacteremia caused by Gram-negative ESKAPE bacilli during the COVID-19 pandemic and characterize the clinical and microbiological findings including antimicrobial resistance. A total of 115 Gram-negative ESKAPE isolates were collected from patients with nosocomial bacteremia (18% of the total bacteremias) in a tertiary care center in Mexico City from February 2020 to January 2021. These isolates were more frequently derived from the Respiratory Diseases Ward (27), followed by the Neurosurgery (12), Intensive Care Unit (11), Internal Medicine (11), and Infectious Diseases Unit (7). The most frequently isolated bacteria were *Acinetobacter baumannii* (34%), followed by *Klebsiella pneumoniae* (28%), *Pseudomonas aeruginosa* (23%) and *Enterobacter* spp (16%). *A. baumannii* showed the highest levels of multidrug-resistance (100%), followed by *K. pneumoniae* (87%), *Enterobacter* spp (34%) and *P. aeruginosa* (20%). The $bla_{CTX-M-15}$ and $bla_{TEM-1}$ genes were identified in all beta-lactam-resistant *K. pneumoniae* (27), while $bla_{TEM-1}$ was found in 84.6% (33/39) of *A. baumannii* isolates. The carbapenemase gene $bla_{OXA-398}$ was predominant among carbapenem-resistant *A. baumannii* (74%, 29/39) and $bla_{OXA-24}$ was detected in four isolates. One *P. aeruginosa* isolate was $bla_{VIM-2}$ gene carrier, while two *K. pneumoniae* and one *Enterobacter* spp were $bla_{NDM}$ gene carriers. Among colistin-resistant isolates *mcr-1* gene was not detected. Clonal diversity was observed in *K. pneumoniae*, *P. aeruginosa* and *Enterobacter* spp. Two outbreaks caused by *A. baumannii* ST208 and ST369 were detected, both belonging to the clonal complex CC92 and IC2. *A. baumannii* was associated with a death rate of 72% (28/32), most of them (86%, 24/28) extensively drug-resistant or pandrug-resistant isolates, mainly in patients with COVID-19 (86%, 24/28) in the Respiratory Diseases Ward. *A. baumannii* isolates had a higher mortality rate (72%), which was higher in patients with

COVID-19. There was no statistically significant association between the multidrug-resistant profile in Gram-negative ESKAPE bacilli and COVID-19 disease. The results point to the important role of multidrug-resistant Gram-negative ESKAPE bacteria causing bacteremia in nosocomial settings before and during the COVID-19 epidemic. Additionally, we were unable to identify a local impact of the COVID-19 pandemic on antimicrobial resistance rates, at least in the short term.

## INTRODUCTION

Antimicrobial resistance (AMR) is a serious global threat for public health. A group of bacteria capable of escaping the action of antibiotics has emerged inside hospitals and since 2008 it has been called the ESKAPE group (*Rice, 2008*). ESKAPE group includes Gram-positive bacteria such as *Staphylococcus aureus*, *Enterococcus faecium* and *Enterococcus faecalis* and Gram-negative bacilli such as *Klebsiella pneumoniae*, *Enterobacter* spp, *Acinetobacter baumannii* and *Pseudomonas aeruginosa*. These last four pathogens are on a list that the World Health Organization (WHO) includes in the most critical group of all, multidrug resistant bacteria that pose a particular threat in hospitals (*WHO, 2017*).

The mechanisms of antimicrobial resistance in Gram-negative bacilli include (i) acquisition of enzymes that modify or destroy antibiotics, (ii) acquisition of enzymes that alter bacterial antibiotic targets, and (iii) acquisition of mutations in bacterial targets that alter antibiotic efficacy or uptake (*Miller, 2016*).

For more than a decade, the importance and need to monitor the frequency of bacteria as the cause of infections within hospitals has been pointed out, as well as the use of typing methods such as pulsed field gel electrophoresis to identify the presence of clones that can cause outbreaks and spread successfully in a hospital environment.

Coronavirus disease (COVID-19) is an infectious disease caused by the SARS-CoV-2 virus. Most people infected with SARS-CoV-2 will experience mild to moderate disease and will not require hospitalization or specialized treatment. However, some individuals can develop a severe acute respiratory syndrome, with serious complications at any age and can die as a result, especially those with underlying illness or morbidities. Coronavirus disease (COVID-19) has had a major impact on healthcare systems and on the incidence of infections caused by multidrug-resistant bacteria (*Imoto et al., 2022*).

Previous data on the incidence of ESKAPE pathogens in a hospital in Monterrey, Mexico showed that the main pathogens isolated in the intensive care unit were *A. baumannii* and *P. aeruginosa* (*Llaca-Díaz et al., 2012*). However, in Mexico there is limited information of bacterial infections in hospitalized patients during the COVID-19 pandemic (*Fernández-García et al., 2022*). Given the importance of conducting studies on the incidence of these pathogens as coinfections of pandemic diseases, its association with outbreaks and mortality, we conducted a study to describe the incidence of bacteremia caused by

Gram-negative ESKAPE bacteria during the COVID-19 pandemic in a public hospital in Mexico City, detailing the clinical and microbiological characteristics, and the pattern of AMR of the isolates.

## MATERIAL AND METHODS

### Clinical setting

Samples were collected at Hospital General de México (HGM) Dr. Eduardo Liceaga from February 2020 to January 2021. HGM is a 1,332-bed tertiary-care teaching hospital in Mexico City, México that serves uninsured adult and pediatric population. Bacterial pathogens were isolated from consecutively collected blood cultures from patients diagnosed with nosocomial bacteremia defined by Infectious Diseases Unit physicians according to criteria published by the Centers for Disease Control (*Horan, Andrus & Dudeck , 2008*). This study was evaluated and approved by the Institutional Research and Ethics Committee project number DI720/405/03/10; 27 February 2020.

### Bacterial isolation

Non-duplicate isolates of *A. baumannii, P. aeruginosa, K. pneumoniae* and *Enterobacter* spp were obtained from blood cultures of patients as part of routine care, identified, and handled anonymously. Bacterial identification was carried out by automated VITEK® 2 system (bioMerieux, Marcy-l'Etoile, France) and stored in Luria Bertani (LB) broth (Difco, BD Biosciences, Franklin Lakes, NJ, United States) with 20% glycerol (Sigma Aldrich, St. Louis, MO, United States) at −70 °C.

### Antimicrobial susceptibility testing

Antimicrobial susceptibility testing and minimum inhibitory concentrations (MIC) were performed by the automated VITEK®-2 System. Microdilution broth method was used to determinate MICs of colistin according to Clinical and Laboratory Standards Institute guidelines (CLSI) (*Melvin et al., 2020*) using *Escherichia coli* ATCC 25922 as quality control strain.

Extended-spectrum beta-lactamases (ESBLs) production in *K. pneumoniae* was confirmed using ceftazidime and cefotaxime disks with or without clavulanic acid (*Melvin et al., 2020*). Metallo-beta-lactamases (MBLs) production was confirmed using meropenem and imipenem disks with or without EDTA (*Calvo et al., 2011*). *Escherichia coli* ATCC 25922 and *Klebsiella pneumoniae* ATCC 700603 were used as quality control strains. Using the criteria of the Latin American Antimicrobial Resistance Surveillance Network (ReLAVRA), the strains were grouped into four different groups: non-multi-drug-resistant (non-MDR), multidrug-resistant (MDR), extensively drug-resistant (XDR) and pandrug-resistant (PDR) (*Jiménez Pearson et al., 2019*).

### Genotypic identification of ESBLs, carbapenemases and mcr-1

The presence of genes that encode beta-lactamases was screened by PCR. The primers used to detect ESBLs genes ($bla_{TEM}$, $bla_{CTX-M}$, and $bla_{SHV}$), MBLs genes ($bla_{VIM}$, $bla_{IMP}$ y $bla_{NDM}$), OXA genes ($bla_{OXA-23}$, $bla_{OXA-24}$, $bla_{OXA-48}$ y $bla_{OXA-58}$) and $bla_{KPC}$ gene were

**Table 1  Primers used for PCR amplification of *β*-lactamase and *mcr-1* genes.**

| Gene | Primer sequence (5′–>3′) | T° annealing (° C) | Amplicon length | Reference |
|---|---|---|---|---|
| $bla_{TEM}$ | F: ATGAGTATTCAACATTTTCG<br>R: TTACCAATGCTTAATCAGTGAG | 55 °C | 861 bp | *Alcántar-Curiel et al. (2019b)* |
| $bla_{SHV}$ | F: ATGCGTTATATTCGCCTGTGTATT<br>R: TTAGCGTTGCCAGTGCTCGATC | 60 °C | 861 bp | *Alcántar-Curiel et al. (2019b)* |
| $bla_{CTX-M}$ | F: CGCTTTGCGATGTGCAG<br>R: ACCGCGATATCGTTGGT | 52 °C | 550 bp | *Alcántar-Curiel et al. (2019b)* |
| $bla_{KPC}$ | F: TCACTGTATCGCCGTCTAGTTCTG<br>R: TTACTGCCCGTTGACGCCCAATC | 58 °C | 875 bp | *Alcántar-Curiel et al. (2019b)* |
| $bla_{VIM}$ | F: GAGTGGTGAGTATCCGACAGTCAACGAAAT<br>R: AGAGTCCTTCTAGAGAATGCGTGGGAATCT | 58 °C | 389 bp | *Alcántar-Curiel et al. (2019b)* |
| $bla_{IMP}$ | F: GCATTGCTACCGCAGCAGAGTCTTTG<br>R: GCTCTAATGTAAGTTTCAAGAGTGATGC | 58 °C | 647 bp | *Alcántar-Curiel et al. (2019a)* |
| $bla_{NDM}$ | F: GTCTGGCAGCACACTTCCTATCTC<br>R: GTAGTGCTCAGTGTCGGCATCACC | 58 °C | 516 pb | *Alcántar-Curiel et al. (2019b)* |
| $bla_{OXA-23}$ | F: GATCGGATTGGAGAACCAGA<br>R: ATTTCTGACCGCATTTCCAT | 52 °C | 501 bp | *Woodford et al. (2006)* |
| $bla_{OXA-24}$ | F: GGTTAGTTGGCCCCCTTAAA<br>R: AGTTGAGCGAAAAGGGGATT | 52 °C | 246 bp | *Woodford et al. (2006)* |
| $bla_{OXA-48}$ | F: GCGTGGTTAAGGATGAACAC<br>R: CATCAAGTTCAACCCAACCG | 58 °C | 438 bp | *Poirel et al. (2011)* |
| $bla_{OXA-51}$ | F: ATGAACATTMAARCRCTCTTACTTA<br>R: CTATAAAATACCTAATTMTTCTAA | 50° C | 825 bp | *Alcántar-Curiel et al. (2019b)* |
| $bla_{OXA-58}$ | F: AAGTATTGGGGCTTGTGCTG<br>R: CCCCTCTGCGCTCTACATAC | 52 °C | 599 bp | *Woodford et al. (2006)* |
| *mcr-1* | F: CGGTCAGTCCGTTTGTTC<br>R: CTTGGTCGGTCTGTAGGG | 45° C | 309 bp | *Liu et al. (2016)* |

previously reported (Table 1). To identify mobile resistance gene associated with colistin resistance, the presence of *mcr-1* was analyzed using *mcr*-1 primers (*Liu et al., 2016*). PCR was performed using the GoTaq Green Master Mix Kit® (Promega, Madison, WI, United States) and amplification conditions described previously (*Alcántar-Curiel et al., 2019b*). Amplified fragments were subjected to Sanger nucleotide sequencing at the Instituto de Biotecnología, Universidad Nacional Autónoma de México. Nucleotide sequence analysis was performed using the BLASTx program of the National Center for Biotechnology Information (http://www.ncbi.nlm.nih.gov/blast).

**Detection of the efflux pumps phenotype**

To determine whether carbapenem resistance was associated with efflux pumps, the phenotypic expression of efflux pumps was determined in carbapenem-resistant isolates. Mueller-Hinton agar plates with meropenem or imipenem double serial dilutions (0.125–256 mg/mL) in the presence or absence of 20 mg/L carbonyl cyanide 3-chlorophenylhydrazone (CCCP) (Sigma, St. Louis, MO, USA), an efflux pump inhibitor were used (*Osei Sekyere, & Amoako, 2017*). Positive phenotypic detection of an efflux

pump was defined two-fold MIC reduction of antibiotics in the presence of the efflux pump inhibitor (*Osei Sekyere, & Amoako, 2017*).

## Pulsed-field gel electrophoresis (PFGE)

The clonal relatedness between the isolates was determined by PFGE as described previously (*Alcántar-Curiel et al., 2019b*). Genomic DNA was digested with *Apa*I for *A. baumannii*, *Spe*I for *P. aeruginosa* and *Xba*I for *Enterobacter* spp and *K. pneumoniae* overnight at 25 °C, 37 °C and 37 °C respectively, followed by PFGE using Gene Path System (Bio-Rad, Hercules, CA, USA). Classification of all clinical isolates into clones was determined by Tenover criteria (*Tenover et al., 1995*).

## Typing of multilocus sequences

Multilocus sequence typing (MLST) was performed in representative *A. baumannii* isolates following the Oxford scheme for *A. baumannii*. The housekeeping genes *gltA, gyrB, gdhB, recA, cpn60, gpi,* and *rpoD* were amplified by PCR and sequenced as previously described (*Bartual et al., 2005*). Allelic profiles and sequence types (ST) were identified using the BIGSdb software from the PubMLST.org website (*Jolley, Bray & Maiden, 2018*). To identify the clonal complex (CC) and visualize the evolutionary relationships between the isolates, the Phyloviz 2.0 program that generates eBURST and a neighbor joining diagram were used (*Sepp et al., 2019*).

## Statistical analyses

The independence analysis of the multidrug resistance levels of the isolates from COVID-19 *versus* non-COVID-19 patients was performed with the bilateral Fisher's Exact test for 2 × 2 contingency tables, using R statistical software. A $P < 0.05$ value was considered statistically significant.

# RESULTS

## Local setting

At the beginning of the study during February to March of 2020 (Period 1), Hospital General de México Dr. Eduardo Liceaga provided healthcare to patients with a whole range of diagnoses and treatment in 48 medical specialties. Given the number of COVID-19 cases during the first wave in the country a large number of patients required hospitalization in the public health system where only a limited number of beds were available. Therefore, health authorities decided turning the institution into a hybrid hospital to dedicate it primarily for SARS-CoV-2 management in the following wards: Respiratory Diseases Ward, Infectious Diseases Unit, Intensive Care Unit, Emergency Department (attention mixed to 50%) and Surgical Tower from April to August 2020 (Period 2, first wave). Afterwards, during the period from September to November 2020 (Period 3), the emergency subsided, and the hospital admitted all types of patients again, including COVID-19. Towards the end of the study, December 2020 to January 2021 (Period 4, second wave), the hospital returned to the hybrid system. During the study period, the hospital provided medical care for uninsured patients residing in all states in the country, and reported approximately 22,000 annual

admissions, (49% less than the annual admissions (42,378) reported in 2019), of which 2,401 (10.9%) had health care-associated infections (HAIs) and of these 638 (26.57%) corresponded to nosocomial bacteremia (*Secretaria de Salud, 2020*).

## Clinical isolates

In the group of nosocomial bacteremias, 115/638 (18%) were caused by Gram-negative bacteria of the ESKAPE group, only one case with coinfection caused by *A. baumannii* and *P. aeruginosa* (0.8%), respectively. The number of isolates was distributed as follows: *A. baumannii* 39/115 (34%), 32 (82%) in COVID-19 patients; *K. pneumoniae* 32/115 (28%), 6 (19%) in COVID-19 patients; *P. aeruginosa* 26/115 (23%), 6 (23%) in COVID-19 patients; *Enterobacter* spp 18/115 (16%), 6 (33%) in COVID-19 patients. September and November concentrated the highest number of isolates (21) (Fig. 1A). The Respiratory Diseases Ward reported the highest number of isolates (27), mainly *A. baumannii* (18), followed by the Neurosurgery (12), Intensive Care Unit (11), Internal Medicine (11), and Infectious Diseases Unit (7) (Fig. 1B). The death rate in patients with *A. baumannii* bacteremia was 72% (28 of 32 patients), which was higher 24/28 (86%) in patients with COVID-19. Among patients with *P. aeruginosa* and *Enterobacter* spp bacteremia both showed a similar death rate of 42% (11 of 26 patients) and 44% (eight of 18 patients), in 55% (6/11) and 50% (4/4) patients with COVID-19 respectively. In the case of *K. pneumoniae* it was 25% (eight of 32 patients) and was lower 2/6 (33%) in patients with COVID-19.

## Antibiotic susceptibility pattern

*A. baumannii* showed the highest levels of AMR of the 4 species studied with 100% resistance to beta-lactam antibiotics, in addition to ciprofloxacin, and 85% (33/39) resistance to gentamicin. Only colistin remained a viable option for treatment, and the usefulness of even this "last resort" antibiotic (Reserve, in the WHO AWaRe classification) was compromised, with 26% (10/39) of resistant *A. baumannii* isolates (Table 2). The proportions of AMR in these isolates were 15% MDR, 67% XDR and 18% PDR. Significantly, a large majority of patients who died had *A. baumannii* XDR or PDR bacteremia (24/28, 86%). In *K. pneumoniae* resistance to beta-lactam antibiotics was 84% (27/32) for 3rd and 4th generation cephalosporins, 6% (2/32) for carbapenems, 66% (21/32) for ciprofloxacin and 62% (20/32) for colistin (Table 2). These isolates were 81% MDR, 6% XDR, 0% PDR and 13% Non-MDR.

*P. aeruginosa* was the bacteria with the lowest percentages of AMR (none higher than 30%), 5/26 (19%) for meropenem and 6/26 (23%) for ciprofloxacin (Table 2). The resistance phenotype for these isolates was 8% MDR, 12% XDR, 0% PDR and 80% Non-MDR. In *Enterobacter* spp resistance to beta-lactam antibiotics was 56% for to 3rd generation cephalosporins, 33% for 4th generation, and 1/18 (6%) to meropenem and 2/18 (11%) for ertapenem, while colistin resistance was 44% (Table 2). It is important to point out that although colistin resistance levels in *K. pneumoniae* and *Enterobacter* spp were high, there are no previous data in our hospital to compare. The resistance phenotype in these isolates was 28% MDR, 6% XDR, 0% PDR and 66% Non-MDR.

The analysis of antimicrobial susceptibility between first and second waves of COVID-19 where there was conversion of exclusive wards for the care of COVID-19 patients

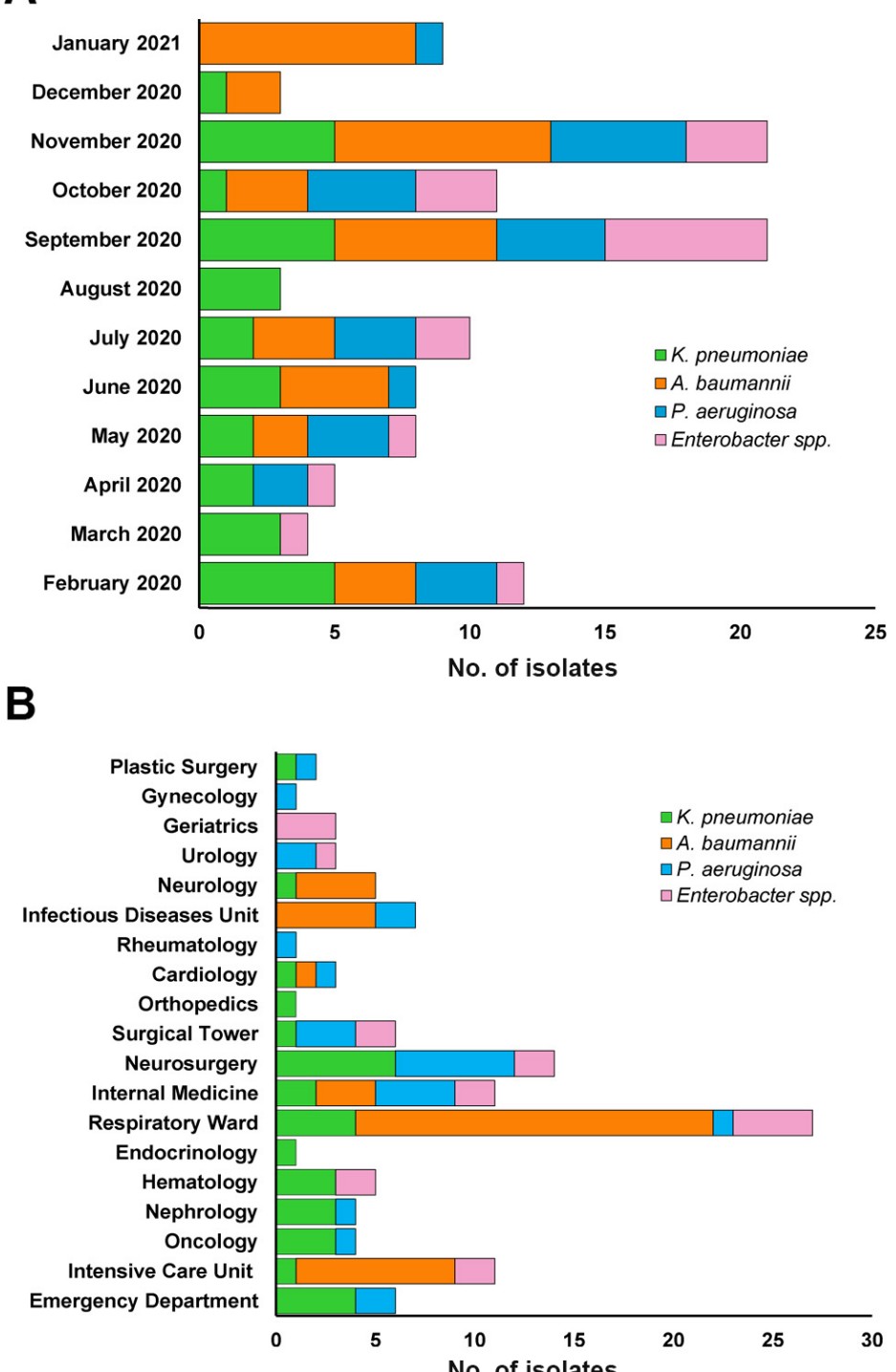

**Figure 1** **Distribution of Gram-negative bacteria causing bacteremia during February 2020 to January 2021.** (A) Monthly distribution. (B) Distribution by hospital service.

and periods 1 and 3 of reception of patients with all types of diseases showed that for *A. baumannii* isolates antimicrobial resistance was high in all four periods of the study for quinolones, cephalosporins, beta-lactams in combination with agents, aminoglycosides, and carbapenems (Fig. 2A). With the exception of gentamicin, where a gradual decrease from 100% intermediate susceptibility was observed in period 1 to 90–100% resistant in periods 2, 3, and 4, and for colistin where a gradual variation from 100% intermediate susceptibility was observed in periods 1 and 4 to 60–70% intermediate susceptibility in periods 2 and 3. Regarding *K. pneumoniae* isolates, these were highly resistant to cephalosporins and beta-lactams in combination with agents in the four periods, with less than 20% susceptibility for periods 2 and 3 and for colistin resistance was high in periods 2 and 3 compared to 100% intermediate susceptibility in periods 1 and 4 (Fig. 2B). In *P. aeruginosa*, high susceptibility to most antibiotics was observed in the four periods, except for ceftazidime and cefepime, which were 100% susceptible in period 2 and decreased to 50% in period 4; for ciprofloxacin, a 45% susceptibility was observed in period 1, which increased to 100% in period 4; for colistin it was 100% intermediate susceptible in periods 1 and 3 and <29% resistant in periods 2 and 4 (Fig. 2C). *Enterobacter* spp isolates showed a sustained increase in resistance to cephalosporin, carbapenems, aminoglycosides and colistin from period 1 to 3 (Fig. 2).

## ESBLs, MBLs and mcr-1 production

In *K. pneumoniae*, 27 isolates resistant to beta-lactams were ESBL producers and all carried the $bla_{CTX-M-15}$ (ID GenBank: AGE61862.1) and $bla_{TEM-1}$ genes (ID GenBank: ALJ57215.1). In *A. baumannii*, 33 of the 39 isolates were carriers of the $bla_{TEM-1}$ gene, but none had the $bla_{SHV}$ or $bla_{CTX-M}$ genes.

All isolates resistant to carbapenems were positive in the confirmatory test to produce MBLs; 39 *A. baumannii*, seven *P. aeruginosa*, two *K. pneumoniae* and one *Enterobacter* spp. The PCR results showed that 29 *A. baumannii* were carriers of $bla_{OXA-398}$ (ID GenBank: WP-063862757.1) and 4 isolates of $bla_{OXA-24}$ (ID GenBank: AXG65603.1). In *P. aeruginosa*, only one isolate was a carrier of the $bla_{VIM-2}$ gene (ID GenBank: AHY39275.1), while the 2 *K. pneumoniae* and 1 *Enterobacter* spp carried the $bla_{NDM}$ gene (ID GenBank: QID22101.1). No isolates carried the $bla_{KPC}$, $bla_{IMP}$, $bla_{OXA-48}$, or $bla_{OXA-58}$ genes. Among colistin-resistant isolates, 20 *K. pneumoniae*, 10 *A. baumannii*, eight *Enterobacter* spp and three *P. aeruginosa*, *mcr-1* gene was not detected.

## Efflux pumps phenotype

In this study *A. baumannii* showed the highest resistance profile, including carbapenems, which are the antibiotics of last choice for the treatment of MDR bacterial infections. The efflux pump phenotype was performed in the 39 isolates of *A. baumannii* resistant to carbapenems, however, none of the isolates were positive. These results confirm that this mechanism is poorly associated with resistance to carbapenems in *A. baumannii*.

## Genotyping of bacterial isolates

All isolates were genotyped by the PFGE technique, the different electrophoretic profiles or pulsetypes received a key in numerical order and the first initial letter of the bacteria.

**Table 2  Minimum Inhibitory Concentration data and antimicrobial susceptibility of Gram-negative bacteria from ESKAPE group.**

| Drug class | AB | Klebsiella pneumoniae n = 32 | | | | | Enterobacter spp n = 18 | | | | | Acinetobacter baumannii n = 39 | | | | | Pseudomonas aeruginosa n = 26 | | | | |
|---|---|---|---|---|---|---|---|---|---|---|---|---|---|---|---|---|---|---|---|---|---|
| | | MIC (μg/ml) | | | MIC50 μg/ml | MIC90 μg/ml | MIC (μg/ml) | | | MIC50 μg/ml | MIC90 μg/ml | MIC (μg/ml) | | | MIC50 μg/ml | MIC90 μg/ml | MIC (μg/ml) | | | MIC50 μg/ml | MIC90 μg/ml |
| | | S | I | R | | | S | I | R | | | S | I | R | | | S | I | R | | |
| **combination agents** | SAM | 5/32 15.6% | 2/32 6.3% | 25/32 78.1% | ≥32 | ≥32 | IR | IR | IR | IR | IR | 0 | 3/3 7.7% | 36/39 92.3% | ≥64/32 | ≥64/32 | IR | IR | IR | IR | IR |
| | TZP | ND | ND | ND | ND | ND | ND | ND | ND | ND | ND | 0 | 0 | 39/39 100% | ≥128/4 | ≥128/4 | ND | ND | ND | ND | ND |
| **Cephems** | FEP | 21/32 65.6% | 4/32 12.5% | 7/32 21.9% | 2 | ≥64 | 12/18 66.7% | 0 | 6/18 33.3% | 1 | ≥64 | 0 | 0 | 39/39 100% | ≥64 | ≥64 | 19/26 73% | 2/26 7.7% | 5/26 19.2% | 2 | ≥64 |
| | CTX | 5/32 16.6% | 0 | 27/32 84.4% | ≥64 | ≥64 | 8/18 44.4% | 0 | 10/18 55.6% | 8 | ≥64 | 0 | 0 | 39/39 100% | ≥64 | ≥64 | IR | IR | IR | IR | IR |
| | CRO | 5/32 16.6% | 0 | 27/32 84.4% | ≥64 | ≥64 | 8/18 44.4% | 0 | 10/18 55.6% | 8 | ≥64 | 0 | 1/39 2.6% | 38/39 97.4% | ≥64 | ≥64 | IR | IR | IR | IR | IR |
| | CFX | 5/32 16.6% | 0 | 27/32 84.4% | ≥64 | ≥64 | 3/18 16.7% | 5/18 27.7% | 10/18 55.6% | 32 | ≥64 | NA | NA | NA | NA | NA | NA | NA | NA | NA | NA |
| | CAZ | ND | ND | ND | ND | ND | ND | ND | ND | ND | ND | 0 | 0 | 39/39 100% | ≥64 | ≥64 | 19/26 73% | 2/26 7.7% | 5/26 19.2% | 4 | ≥64 |
| **Carbapenems** | ETP | 30/32 93.8% | 0 | 2/32 6.2% | 0.5 | 0.5 | 15/18 83.3% | 1/18 5.6% | 2/18 11.1% | 0.5 | 1 | NA | NA | NA | NA | NA | IR | IR | IR | IR | IR |
| | DOR | ND | ND | ND | ND | ND | ND | ND | ND | ND | ND | 0 | 0 | 39/39 100% | ≥8 | ≥8 | ND | ND | ND | ND | ND |
| | IPM | ND | ND | ND | ND | ND | ND | ND | ND | ND | ND | 0 | 0 | 39/39 100% | ≥16 | ≥16 | ND | ND | ND | ND | ND |
| | MEM | 30/32 93.8% | 0 | 2/32 6.2% | 0.25 | 0.25 | 17/18 94.4% | 0 | 1/18 5.6% | ≤0.25 | ≤0.25 | 0 | 0 | 39/39 100% | ≥16 | ≥16 | 17/25 68% | 3/25 12% | 5/25 20% | 1 | ≥16 |
| **Lipopetides** | COL | 0 | 12/32 37.5% | 20/32 62.5% | 8 | 32 | 0 | 10/18 55.6% | 8/18 44.4% | 2 | 32 | 0 | 29/39 74.4% | 10/39 26.6% | 2 | 8 | 0 | 23/26 88.5% | 3/26 11.5% | 1 | 4 |
| **Aminoglycosides** | GEN | 17/32 53.1% | 0 | 15/32 46.9% | ≤1 | 16 | 14/18 77.7% | 0 | 4/18 22.2% | ≤1 | ≥16 | 0 | 6/39 15.4% | 33/39 84.6 | ≥16 | ≥16 | 21/26 80.8% | 0 | 5/26 19.2% | 2 | ≥16 |
| | AMK | 30/32 93.8% | 0 | 2/32 6.2% | 2 | 8 | 16/18 88.9% | 1/18 5.6% | 1/18 55.6% | ≤2 | 16 | ND | ND | ND | ND | ND | 23/26 88.5% | 0 | 3/26 11.5% | ≤2 | ≥64 |
| **Quinolones** | CIP | 8/32 25% | 3/32 9.4% | 21/32 65.6%- | ≥4 | ≥4 | 12/18 66.7% | 0 | 5/18 27.8% | ≤0.25 | ≥2 | 0 | 0 | 39/39 100% | ≥4 | ≥4 | 20/26 77% | 0 | 6/26 23% | ≤0.25 | ≥4 |

**Notes.**

AB, antibiotic; MIC, Minimum Inhibitory Concentration; I, intermediate susceptibility; R, resistant; S, susceptible; SAM, ampicillin-sulbactam; TZP, piperacillin/tazobactam; FEP, cefepime; CTX, cefotaxime; CRO, ceftriaxone; CFX, cefuroxime; CAZ, ceftazidime; ETP, ertapenem; DOR, doripenem; IPM, imipenem; MEM, meropenem; COL, colistin; GEN, gentamicin; AMK, amikacin; CIP, ciprofloxacin; IR, íntrinsic resistance; ND, not determined; NA, Does not.

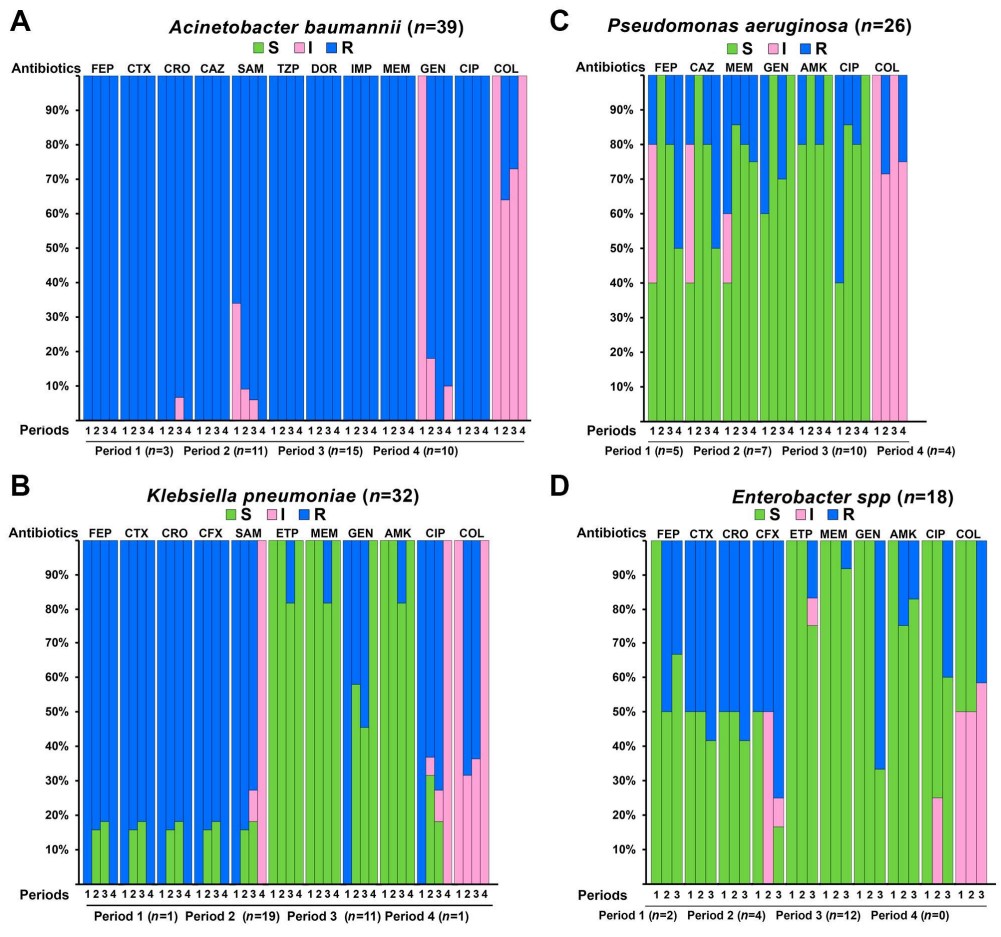

**Figure 2** **Antimicrobial susceptibility profiles of Gram-negative bacteria of the ESKAPE group during the different periods with and without conversion of exclusive services for the care of COVID-19 patients.** I, intermediate resistant; R, resistant; S, susceptible; SAM, ampicillin-sulbactam; TZP, piperacillin/-tazobactam; FEP, cefepime; CTX, cefotaxime; CRO, ceftriaxone; CFX, cefuroxime; CAZ, ceftazidime; ETP, ertapenem; DOR, doripenem; IPM, imipenem; MEM, meropenem; COL, colistin; GEN, gentamicin; AMK, amikacin; CIP, ciprofloxacin. Period 1: February to March of 2020, Period 2: April to August 2020, Period 3: September to November 2020, Period 4: December 2020 to January 2021.

Clonal relatedness analysis identified 16 different pulsetypes in 39 *A. baumannii* isolates (A1 to A16) (Fig. 3A). During the period between the first and second COVID-19 waves in which the hospital re-admitted all types of patients (September to November 2020), two nosocomial outbreaks were identified; the first was caused by 11 isolates of pulsetype 4 in patients with COVID-19 during September–November in the Respiratory Diseases Ward. The second outbreak was caused by three isolates of pulsetype 7 also in COVID-19 patients in the Infectious Diseases Unit during November–December 2020. In *K. pneumoniae* electrophoretic profiles were obtained in 29/32 isolates (three isolates were not typable) and were grouped into 25 different pulsetypes (K1 to K25) (Fig. 3B). In *P. aeruginosa* 21/26 isolates were grouped in 20 different pulsetypes (P1 to P20), five isolates were not typable (Fig. 3C). The 17/18 isolates of *Enterobacter* spp electrophoretic profiles were grouped in

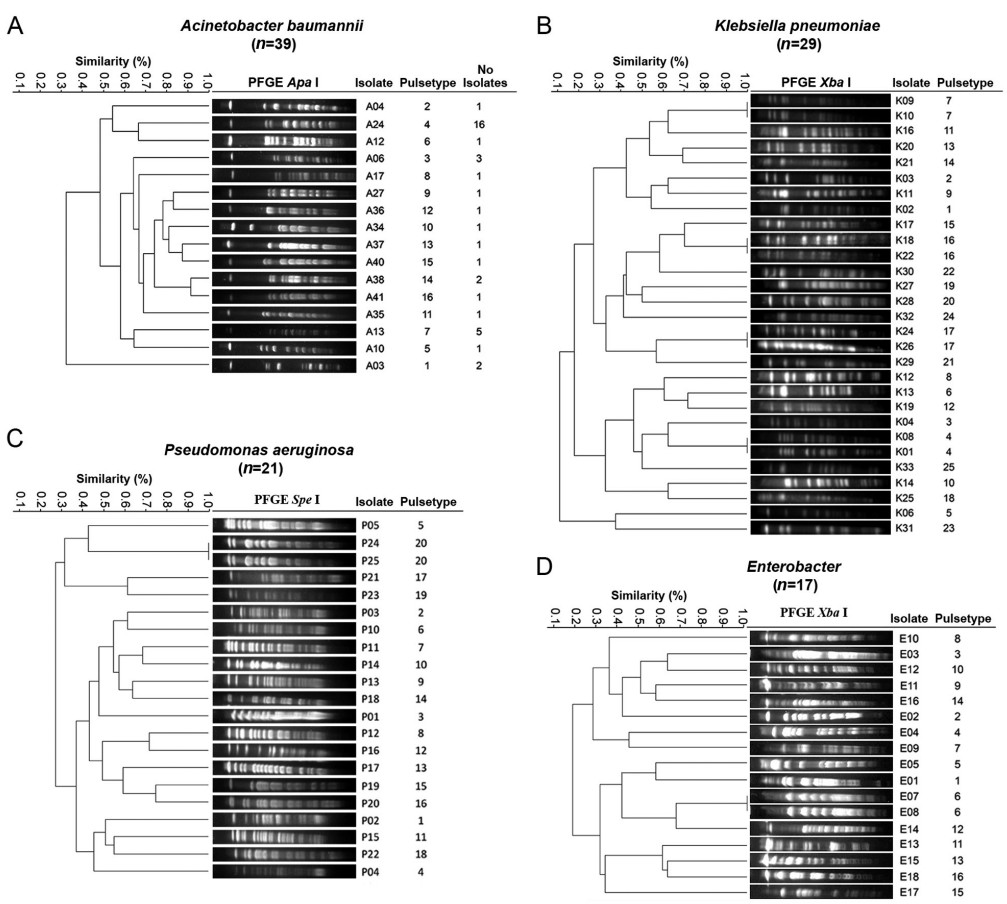

**Figure 3** **Dendrogram generated from the analysis of PFGE patterns of Gram-negative bacteria isolates from a tertiary care Mexican hospital.** (A) A total of 39 isolates of *A. baumannii* were grouped in 16 different pulsetypes, one PFGE representative profile by pulsetype was used to construct the dendrogram. (B) A total of 29 isolates of *K. pneumoniae* e were grouped in 25 different pulsetypes. (C) A total of 21 of *P. aeruginosa* were grouped in 20 different pulsetypes. (D) A total of 17 isolates of *Enterobacter spp* were grouped in 16 pulsetypes. The dashed line represents the 85% similarity level used in the cluster designation.

16 different pulsetypes (E1 to E16), one isolate was not typable (Fig. 3D). Analysis of these findings revealed that *K. pneumoniae, P. aeruginosa* and *Enterobacter* spp showed clonal diversity.

## MLST sequence types among *A. baumannii* isolates causing nosocomial outbreaks

MLST analysis of 5/11 isolates pulsetype 4 involved in the first outbreak showed that all these belong to ST369. The three isolates pulsetypes 7 caused the second outbreak belonged to ST208. These two STs belonged to clonal complex CC92. These results confirm the association of these isolates with the production of two outbreaks.

**Table 3  Fisher's Exact test results for multidrug resistance of isolates from COVID and non-COVID patients.**

| Bacteria (n) | Patient | No. isolates (%) | Non-MDR (%) | Resistance (MDR+XDR+PDR) MDR (%) XDR (%) PDR (%) | | | P value |
|---|---|---|---|---|---|---|---|
| | | | | MDR (%) | XDR (%) | PDR (%) | |
| A. baumannii | COVID | 32 (80%) | 0 (0%) | 3 (9%) | 22 (69%) | 7 (22%) | |
| n = 39 | NON-COVID | 7 (20%) | 0 (0%) | 3 (43%) | 4 (57%) | 0 (0%) | 1.0 |
| K. pneumoniae | COVID | 6 (23%) | 0 (0%) | 4 (67%) | 2 (33%) | 0 (0%) | |
| n = 32 | NON-COVID | 26 (77%) | 4 (15%) | 22 (85%) | 0 (0%) | 0 (0%) | 0.5662 |
| P. aeruginosa | COVID | 6 (22%) | 4 (67%) | 0 (0%) | 2 (33%) | 0 (0%) | |
| n = 26 | NON-COVID | 20 (78%) | 17 (85%) | 2 (10%) | 1 (5%) | 0 (0%) | 0.5581 |
| Enterobacter spp | COVID | 6 (35%) | 4 (67%) | 2 (33%) | 0 (0%) | 0 (0%) | |
| n = 18 | NON-COVID | 12 (65%) | 8 (67%) | 3 (25%) | 1 (8%) | 0 (0%) | 1.0 |

**Notes.**

Sum of percentages of MDR, XDR and PDR of each bacterial species in the group of patients with COVID-19 was compared with those obtained in the group of non-COVID-19 patients.

MDR, multidrug-resistant; XDR, extensively drug-resistant; PDR, pandrug-resistant; non-MDR, non-multidrug- resistant.

A $P < 0.05$ value was considered statistically significant.

### Multi-resistance analysis in COVID-19 *vs* non-COVID-19 patients

A total of 32/39 (80%) *A. baumannii* isolates were obtained from COVID-19 patients, all of these showed an MDR profile. All seven (20%) isolates from non-COVID-19 patients also showed an MDR profile, this proportion is expected because the patients with COVID-19 corresponded to severe and critical patients undergoing invasive procedures, with a long hospital stay in intensive care units. Results of the Fisher's Exact test showed that there was no statistically significant association between MDR profile of *A. baumannii* and COVID-19 and non-COVID-19 patient since Fisher's Exact test had a *p*-value of 1 ($P > 0.05$) (Table 3). In *K. pneumoniae* 26/32 (81%) isolates were obtained from non-COVID-19 patients, the vast majority 22/26 (85%) showed an MDR profile. There was no statistically significant association between MDR profile of *K. pneumoniae* and COVID-19 and non-COVID-19 patient since Fisher's Exact test had a *p*-value of 0.5662 ($P > 0.05$) (Table 3). Similar results were observed in *Enterobacter* spp isolates, 33% of isolates from non-COVID-19 patients and 33% of isolates from COVID-19 patients showed an MDR profile. Fisher's Exact test had a *p*-value of 1 (Table 3), showed that there was no statistically significant association ($P > 0.05$) between MDR profiles of *Enterobacter* spp isolates from the two groups of patients. In the case of *P. aeruginosa*, the vast majority, 85% of isolates from non-COVID-19 patients and 67% of isolates from COVID-19 patients were susceptible. Fisher's Exact test had a *p*-value of 0.5581 (Table 3), showed that there no statistically significant association ($P > 0.05$) between MDR profile of *P. aeruginosa* isolates from the two groups of patients.

### DISCUSSION

The COVID-19 epidemic has had and continues to have, a direct and dramatic impact on all socioeconomic aspects worldwide and the health area was the most affected (*Nicola et*

*al., 2020*), Mexico has not been the exception. The large number of hospitalizations due to severe infection with the virus brought was associated with an increase in the number of Healthcare-Associated Infections (HAIs) due to the predisposition of these patients to acquire a secondary infection, and most of these infections are caused by bacteria from the ESKAPE group (*Baker et al., 2022*).

In this study, several wards of the Hospital General de México Dr. Eduardo Liceaga were converted to the exclusive care of COVID-19 patients during the first and second waves of the COVID-19 pandemic was associated with a decrease in the admission of patients with other types of medical or surgical conditions, which explains the decrease in the number of nosocomial infections documented during 2020 by the Epidemiology Department 2,401 compared to 3,163 reported the previous year (according to the internal HAIs database from the Department of Epidemiology in 2019).

A recent study reported bacterial coinfections/superinfections in COVID-19 patients with assisted mechanical ventilation (VAP), where *P. aeruginosa* and *K. pneumoniae* were the most frequent in bronchial aspirate samples (*Mazzariol et al., 2021*). Our investigation showed that *A. baumannii* was the Gram-negative bacteria of the ESKAPE group most frequent in blood cultures, followed by *K. pneumoniae*, *P. aeruginosa*, and *Enterobacter* spp. This data was different from the previous year in the hospital, where *K. pneumoniae* was the most frequent (40%), followed by *A. baumannii* (17%) and *P. aeruginosa* (6%) (*Blancas Reyes, 2019*). These results are consistent with the period of hospital conversion in 2020 in which patients with COVID-19 were mainly admitted and due to their treatment, there was an increase in VAP, many of which led to *A. baumannii* bloodstream infections. In addition, September and November concentrated the highest number of isolates, these months coincide with the period in which the hospital worked in a hybrid mode, caring for both patients with COVID-19 and those with other types of ailments, and in this period all the wards were open and had a greater influx of patients (Fig. 1A). The services where the largest number of Gram-negative bacteria were isolated (Respiratory Diseases Ward, Neurosurgery, Intensive Care, Internal Medicine, and Infectious Diseases) (Fig. 1B), agree with the wards were the largest number of patients with COVID-19 were admitted, except for Neurosurgery. Ours results reveal that *A. baumannii* was associated with a high death rate (72%), most of these deaths were associated with XDR or PDR isolates in COVID-19 patients cared for in the Respiratory Diseases Ward; these data are consistent with those recently reported in a comparative study of the risk of death from secondary bacterial infections between patients with COVID-19 *vs* patients with influenza, where death rates were found to be higher for COVID-19 patients with early infections 42.9% and late infections 66% (*Shafran et al., 2021*).

The AMR results of this study showed that antimicrobial therapy with beta-lactam antibiotics used for the treatment of nosocomial bacteremia is highly compromised in this hospital, since both *K. pneumoniae* and *Enterobacter* spp showed high levels of resistance to 3rd and 4th generation cephalosporins, as well as *A. baumannii* isolates that showed resistance to practically all antibiotics of this family, including carbapenems. It is important to point out that although the overall level of antimicrobial resistance in these isolates did not increase compared to the previous year (according to the internal database from the

Department of Infectious Diseases Unit in 2019), the detection of these strains represents a major problem in the hospital because these organisms can cause serious infections that are difficult to manage and put the patient's life at risk.

*K. pneumoniae* isolates in our study were CTX-M-15, TEM-1 and NDM-1-producers, these results were comparable to those reported in a study of isolates from Hospital Civil de Guadalajara (HCG) in Guadalajara Jalisco, Mexico (*Toledano-Tableros et al., 2021*). The production of NDM-1 in *Enterobacter* spp isolates in this work is consistent with what we reported at the ISSSTE Hospital in Mexico City (*Alcántar-Curiel et al., 2019a*), and the OXA-24 type carbapenemases produced by *A. baumannii* isolates are in agreement with our findings in HCG (*Alcántar-Curiel et al., 2019b*). These results demonstrate the dissemination of these enzymes among beta-lactams-resistant isolates that cause bacteremia and indicate the prevalence and endemic behavior of this family of ESBLs, oxacillinases and carbapenemases in different areas of Mexico. The identification of the OXA-398 enzyme belonging to the OXA-23 family (*Nigro & Hall, 2016*) in clinical isolates of *A. baumannii* is the first report in Mexico and suggests its recent acquisition in clinical isolates.

The carbapenemase-producing isolates with the absence of the carbapenemase genes investigated in our study (six *P. aeruginosa* and six *A. baumannii*), suggest that they could have different carbapenemases than the one already identified, such as SIM (*Lee et al., 2005*) or have other mechanisms such as loss of porins, since they have not overexpression of efflux pumps (*Hassuna et al., 2020*).

In our study, the isolates of *K. pneumoniae* and *Enterobacter* spp were 62.5% and 44.4% resistant to colistin, respectively. These findings showed that colistin is not an option for the treatment of bacteremia caused by these strains. Since susceptibility to carbapenems in Enterobacterales remains high in our setting and due to the high level of resistance observed to other antibiotics, the use of carbapenems should be considered as more effective. Regarding bacteremia caused by *P. aeruginosa* that exerted high levels of susceptibility to various antibiotics, colistin should be considered as one of the last options as a high rate of nephrotoxicity (*Oliota et al., 2019*). *A. baumannii* isolates were resistant to all families of antibiotics, including carbapenems, however, they showed good levels of susceptibility to colistin, and therefore, colistin could be used.

Antibiotic resistance profiles between periods of general hospital (1 and 3) and COVID-19 hospital (2 and 4) showed that *A. baumannii* isolates presented high MDR profiles even before the COVID-19 pandemic and were maintained in the following periods studied (Fig. 2A). The high resistance to cephalosporins, the susceptibility to carbapenems and the intermediate resistance to colistin of *K. pneumoniae* was maintained in the different periods (Fig. 2B). In general, the intermediate susceptibility to cephalosporins, carbapenems, aminoglycosides and fluoroquinolones and the susceptibility to colistin of *P. aeruginosa* was also maintained in the different periods (Fig. 2C). A slight increase in resistance was observed in a sustained manner in the three periods for *Enterobacter* spp which had the lowest number of isolates (Fig. 2D). We believe that our results reflect that during the first year the COVID-19 pandemic had no effect on antimicrobial resistance in the isolates identified in our hospital.

This study has several limitations: it was a single-center study conducted in Mexico, some of the periods studied had very few isolates (*A. baumannii* period 1, *K. pneumoniae* periods 1 and 4, *Enterobacter* spp period 1) or even there were no isolates (*Enterobacter* spp period 4), and that the study only covers the first year of the pandemic. It is important to note that, the study only included Gram-negatives from the ESKAPE group that caused bacteremia, this undoubtedly influenced the low percentage of coinfections observed. Recently it has been reported that the most common bacterial respiratory and bloodstream pathogens that caused coinfections upon hospital admissions of patients with COVID-19 are Gram-positive cocci *Staphylococcus aureus*, *Streptococcu pneumoniae* and that the blood cultures have low yield in hospitalized patients with COVID-19 (*Westblade, Simon & Satlin, 2021*).

The identification of nosocomial outbreaks due to *A. baumannii* pulsetype 7 in the Infectious Diseases Unit and the pulsetype 4 in the Respiratory Diseases Ward (Fig. 3A), warns about the presence of high-risk MDR clones in the wards where critically ill patients are treated, such as those with COVID-19, for which it is necessary to carry out epidemiological surveillance.

The detection of isolates of *A. baumannii* pulsetype 7, ST208, CC92 causing outbreak in Infectious Diseases Unit in our study (Fig. 3A) and the previous detection of this same pulsetype among *A. baumannii* isolates from Hospital Civil de Guadalajara (named pulsetype G, ST208 and CC92) (*Mateo-Estrada et al., 2021*), demonstrate the spread of lineage of *A. baumannii* multidrug-resistant between hospitals in different geographic areas of Mexico in a short period of time. ST208 belonging to the international clone 2 (IC2) is strongly associated with multidrug resistant, including carbapenems, which has caused nosocomial outbreaks worldwide and has recently been reported in Mexico (*Mateo-Estrada et al., 2021*). Isolates causing outbreak in Respiratory Diseases Ward belonged to different lineage, ST369, although this ST also belongs to IC2. Hence, it seems that different lineages can be coexisting within the hospital. From a practical point of view, the distribution of different STs per ward suggests that infection control measures were able to contain the dissemination of these STs between different areas within the same hospital setting.

Although our results showed that MDR Gram-negative ESKAPE bacteria are prevalent in both COVID-19 and non-COVID-19 patients and no association between MDR and COVID-19 was observed, the potential of MDR bacteria to cause infectious complications should be considered in COVID-19 patients due to prolonged hospitalization and immunosuppression (*Bongiovanni et al., 2022*).

## CONCLUSION

Given the importance of epidemiological surveillance of the AMR problem throughout the world, our study demonstrates the important role of MDR Gram-negative ESKAPE bacteria isolates that caused bacteremia in nosocomial settings before and during the COVID-19 pandemic. Our results show that the COVID-19 pandemic did not have a local impact on antimicrobial resistance rates in the short term of this study. However, the presence of MDR isolates in the hospital setting emphasizes the need to update and strengthen antimicrobial

stewardship programs and the importance of maintaining surveillance actions to prevent the spread of MDR bacteria.

## ACKNOWLEDGEMENTS

We thank Infectious Diseases Residents Janeth Carolina Nevarez Luján and Eduardo Alemán Garay from the Hospital General de México Dr. Eduardo Liceaga, the capture of clinical records. The authors are grateful to Tec. SD. Marco Elias Gudiño Zayas from the Universidad Nacional Autónoma de México, for assistance with the graphic design. The authors also thank Dr. Victor Hugo Olmedo Canchola, Coordinador de Programas Académicos de Posgrado, Facultad de Medicina from the Universidad Nacional Autónoma de México for assistance with statistical analysis.

### Funding

This work was supported by the Programa de Apoyo a Proyectos de Investigación e Innovación Tecnológica (UNAM-PAPIIT-DGAPA) grant number IN217721 and Consejo Nacional de Ciencia y Tecnología (CONACyT México) grant number Ciencia de Frontera 2019-171880. MH-C received fellowship (grant number 1035418) from CONACYT for MSc studies. The funders had no role in study design, data collection and analysis, decision to publish, or preparation of the manuscript.

### Grant Disclosures

The following grant information was disclosed by the authors:
Programa de Apoyo a Proyectos de Investigación e Innovación Tecnológica (UNAM-PAPIIT-DGAPA): IN217721.
Consejo Nacional de Ciencia y Tecnología (CONACyT México): Ciencia de Frontera 2019-171880.
CONACYT: 1035418.

### Competing Interests

The authors declare there are no competing interests.

### Author Contributions

- María Dolores Alcántar-Curiel conceived and designed the experiments, analyzed the data, authored or reviewed drafts of the article, and approved the final draft.
- Manuel Huerta-Cedeño conceived and designed the experiments, performed the experiments, analyzed the data, prepared figures and/or tables, and approved the final draft.
- Ma Dolores Jarillo-Quijada analyzed the data, prepared figures and/or tables, and approved the final draft.
- Catalina Gayosso-Vázquez analyzed the data, prepared figures and/or tables, and approved the final draft.

- José Luis Fernández-Vázquez analyzed the data, authored or reviewed drafts of the article, and approved the final draft.
- María Luisa Hernández-Medel performed the experiments, authored or reviewed drafts of the article, and approved the final draft.
- Manuelita Zavala-Pineda performed the experiments, authored or reviewed drafts of the article, and approved the final draft.
- Miguel Ángel Morales-Gil performed the experiments, authored or reviewed drafts of the article, and approved the final draft.
- Verónica Alejandra Hernández-Guzmán performed the experiments, authored or reviewed drafts of the article, and approved the final draft.
- Manuel Ismael Bolaños-Hernández performed the experiments, authored or reviewed drafts of the article, and approved the final draft.
- Silvia Giono-Cerezo conceived and designed the experiments, analyzed the data, authored or reviewed drafts of the article, and approved the final draft.
- José Ignacio Santos-Preciado analyzed the data, authored or reviewed drafts of the article, and approved the final draft.

## Ethics

The following information was supplied relating to ethical approvals (i.e., approving body and any reference numbers):

The Hospital General de México Dr. Eduardo Liceaga granted Ethical approval to carry out the study within its facilities (Ethical Application Ref: DI720/405/03/10; 27 February 2020).

## DNA Deposition

The following information was supplied regarding the deposition of DNA sequences:

Sequence accession numbers are in the Supplemental File.

https://www.ncbi.nlm.nih.gov/genbank/.

## Data Availability

The raw measurements are available in the Supplementary File.

## Supplemental Information

Supplemental information for this article can be found online at http://dx.doi.org/10.7717/peerj.15007#supplemental-information.

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
