# Peer review of "Gram-negative ESKAPE bacteria bloodstream infections in patients during the COVID-19 pandemic"

_PeerJ, doi:10.7717/peerj.15007_

## Round 0.1 · original submission · Minor Revisions

As the authors will realize, a few minor issues suggested by the reviewers need to be resolved before accepting the manuscript for publication.

Reviewer 1 ·

Basic reporting

Alcántar-Curiel et al have presented a useful and relevant data and study on bloodstream infections caused by ESKAPE bacteria during the COVID-19 pandemic. This study is relevant for future preparedness and fight against antibiotic tolerant bacteria. I enjoyed reading the article and recommend its publication in PeerJ.

The article is clear and professional English language is used throughout.

Background/introduction is written clearly and provides context for a wide audience.

The structure of the manuscript conforms to PeerJ standards and raw data has been shared.

Experimental design

The study provides rigorous investigation of an important health problem. The research questions
is well defined and meaningful. I like how the article is written to be accessible to experts as well as someone with general science background. Not a lot of articles are written this way and I enjoyed reading it. Methods section is clearly written and detailed. IBC provided and uncropped gel details also provided as supplemental.

Validity of the findings

Although I enjoyed reading the paper for most part, the authors should provide additional details about 'Statistical analyses' section (see notes below).

Statistical testing needs to be explained better and needs more context. Specifically, was the chi-square test selected? Why not use Fisher’s Exact test, which might be more appropriate for smaller samples? Also, how were the p-values calculated? Software (Excel, Matlab, Python) or tables? I think it would be useful to add these details in the ‘Statistical Analyses’ section. Also, authors should describe the distinction between p-value and P values in this section. Additionally: the chi-square is reported with three decimal levels of accuracy here. Is that necessary? Authors should also report chi-square value used for statistical testing in the methods section.

Additional comments

Minor comments:

116: Missing comma after (non-MDR)
182: ‘bacteria’ instead of ‘bacilli’?
235: What are ‘BLEEs’? Authors should consider giving a more useful title.
246: comma instead of semi-colon here and other places. Please check manuscript for this error.
292: Missing ‘of’ between the ‘p-value’ and the reported number at multiple places in 280-297
296: semi-colon not appropriate here
328-330: The sentence is unclear.

I also have a general question for the authors Do the authors observe co-infections of bacteria in their samples? If not, what might be the reasons for it? Are there studies in literature that have similar observations?

·

Basic reporting

This article is ready to be published as soon as possible.
This article is profound and very clear in its goal. it has a lucid language which is very helpful to understand for a lay man also.
All references are well coordinated, and all placed in the proper positions. The background research is also done very meticulously so nothing to add in there.
All the sections are portioned carefully and well designated. Figures and table are appropriate.
As they are trying to tell the increased no. of ESKAPE bacteria in COVID_19 patients than non-COVID19 patients, their results did justice to that hypothesis.

Experimental design

Experimental design is well-defined, relevant and meaningful. All the needful experiments was there in the article to address this issue.

Validity of the findings

findings are robust and well organized in tables and figures. Conclusions is well stated and linked with original questions.

Additional comments

I will highly recommend the article to be published.

·

Basic reporting

1. The English of this manuscript should be substantially improved. It is not easy to read through the whole manuscript.

Experimental design

1. Materials and Methods section (line number: 130) what threshold and parameters have the author used for nucleotide analysis?
2. On the other hand, the analysis does not include other important gram negative bacteria and Enterobacterales. The lack of this data is especially evident in the identification of relatedness among carbapenems.

Validity of the findings

1. Why authors have not included patients of COVID-19 third wave.
2. I think, authors need to conduct the study in future with more number of isolates which covers the other years of pandemic too.

Additional comments

1. Your introduction needs more details. I suggest you improve your introduction substantially to provide more justification for your study (specifically, you should expand upon the knowledge gap being filled) and key findings.
2. The abstract needs more details as well as, please include weakness of previous studies and your motivations behind the present study.

Minor Comments
1. In the abstract (line number 47) abbreviation was used. This was used for the first time in the text. Do not use abbreviations.
2. Figure 1 change (No. isolates) with (No. of isolates)

---

## Round 0.2 · accepted · Accept

The authors successfully implemented all reviewers' comments. I congratulate the authors for their work.

·

Basic reporting

No Comment

Experimental design

No Comment

Validity of the findings

No Comment

Additional comments

I endorse the current version of manuscript for publication.